# Dissecting the Protective Effect of CD8^+^ T Cells in Response to SARS-CoV-2 mRNA Vaccination and the Potential Link with Lymph Node CD8^+^ T Cells

**DOI:** 10.3390/biology12071035

**Published:** 2023-07-22

**Authors:** Mengfei Chen, Vanessa Venturi, C. Mee Ling Munier

**Affiliations:** The Kirby Institute, UNSW, Sydney, NSW 2052, Australia; mchen@kirby.unsw.edu.au (M.C.); vventuri@kirby.unsw.edu.au (V.V.)

**Keywords:** CD8^+^ T cells, SARS-CoV-2, mRNA vaccination, lymph node

## Abstract

**Simple Summary:**

Coronavirus disease 2019 (COVID-19) is a respiratory disease caused by the novel coronavirus severe acute respiratory syndrome coronavirus 2 (SARS-CoV-2), which has led to millions of deaths globally. The rollout of SARS-CoV-2 vaccines has effectively reduced the morbidity and mortality of COVID-19, with messenger RNA (mRNA)-based vaccines being widely administrated. While neutralizing antibodies are crucial, CD8^+^ T cells induced by the vaccine may also play a significant role in early and long-term protection. This review explores the antiviral function of CD8^+^ T cells and their response to mRNA vaccines, as well as their role in immune protection in the lymph nodes.

**Abstract:**

SARS-CoV-2 vaccines have played a crucial role in effectively reducing COVID-19 disease severity, with a new generation of vaccines that use messenger RNA (mRNA) technology being administered globally. Neutralizing antibodies have featured as the heroes of vaccine-induced immunity. However, vaccine-elicited CD8^+^ T cells may have a significant impact on the early protective effects of the mRNA vaccine, which are evident 12 days after initial vaccination. Vaccine-induced CD8^+^ T cells have been shown to respond to multiple epitopes of SARS-CoV-2 and exhibit polyfunctionality in the periphery at the early stage, even when neutralizing antibodies are scarce. Furthermore, SARS-CoV-2 mRNA vaccines induce diverse subsets of memory CD8^+^ T cells that persist for more than six months following vaccination. However, the protective role of CD8^+^ T cells in response to the SARS-CoV-2 mRNA vaccines remains a topic of debate. In addition, our understanding of CD8^+^ T cells in response to vaccination in the lymph nodes, where they first encounter antigen, is still limited. This review delves into the current knowledge regarding the protective role of polyfunctional CD8^+^ T cells in controlling the virus, the response to SARS-CoV-2 mRNA vaccines, and the contribution to supporting B cell activity and promoting immune protection in the lymph nodes.

## 1. Introduction

Coronavirus disease 2019 (COVID-19), evoked by the severe acute respiratory syndrome coronavirus 2 (SARS-CoV-2), caused a sizeable outbreak in China at the end of 2019 with a high rate of morbidity and mortality. On 11 March 2020, the World Health Organization (WHO) declared this outbreak a global pandemic following the rapid spread of this disease worldwide [1]. As of 17th May 2023, according to the WHO COVID-19 dashboard (https://covid19.who.int, accessed on 17 May 2023), the COVID-19 pandemic has resulted in 766 million confirmed cases and 6.9 million deaths worldwide. The SARS-CoV-2 virus is a member of zoonotic coronaviruses that cause an acute infection in the upper respiratory tract leading to a severe respiratory syndrome [2]. The distinct structure of SARS-CoV-2 enables rapid infection and transmission among individuals. There are four structurally important proteins of SARS-CoV-2: the Nucleocapsid protein (N), Spike glycoprotein (S), Membrane protein (M), and Envelope protein (E) [3,4]. The Spike protein plays a crucial role in infecting host cells as it contains the receptor binding domain that binds to the cognate receptor angiotensin-converting enzyme 2 (ACE2) on the host cell surface, which enables the SARS-CoV-2 virus to enter and replicate within the host cells [4,5]. There are a wide range of symptoms associated with SARS-CoV-2 that vary between individuals [6,7,8]. Some of the most common symptoms include headache, fever, cough, shortness of breath, and sore throat [6,7,8]. However, in severe cases, this virus may invade the nervous and cardiovascular systems, leading to potentially fatal symptoms [9]. 

To combat the COVID-19 pandemic, a number of vaccines against SARS-CoV-2 were developed and rolled out, leading to a significant reduction in morbidity and mortality [10]. Among them, messenger RNA (mRNA)-based vaccines were initially considered favorable due to their rapid onset of action, robust efficacy, and low chance of adverse events [11,12,13]. The remarkable intervention of prime-boost vaccination using mRNA-based vaccines was adopted in many countries. The two mRNA vaccines rolled out were the Pfizer-BioNTech BNT162b2 and Moderna mRNA-1273, which exhibited >94% efficacy in combating the ancestral strain of virus [11,12]. These vaccines use a lipid nanoparticle (LNP) to encapsulate single-stranded mRNA that encodes for the SARS-CoV-2 Spike glycoprotein [14]. The initial vaccines featured mRNA encoding the ancestral strain, referred to as Wuhan-Hu-1 [15]. In addition, the more recent bivalent mRNA vaccines contain mRNA that encodes for both the ancestral strain and one or more Omicron variants [15]. Following vaccination, the LNP encapsulated mRNA is taken up by host antigen presenting cells (APCs) and processed to produce the Spike viral protein, triggering an immune response [16]. Neutralizing antibodies (NAbs) targeting the Spike protein of SARS-CoV-2 are capable of neutralizing the virus, which results in a prophylactic effect from disease [16,17]. Interestingly, scientists have reported that vaccine-induced protection is achieved by 12 days after the prime vaccination [11], even when NAbs were barely detectable [17,18]. Further investigation suggests that vaccine-induced CD8^+^ T cells may substantially contribute to the protective efficacy of the SARS-CoV-2 mRNA vaccine at this early timepoint [19]. 

The involvement of CD8^+^ T cells in virologic control has been demonstrated by numerous studies. CD8^+^ T cells, also known as cytotoxic T lymphocytes (CTL), play a significant role in adaptive immune responses due to their unique cell-killing mechanisms, which induce sterilizing immunity. Over the last decades, there have been many studies of the CD8^+^ T cell response to vaccination in peripheral blood. However, in the context of SARS-CoV-2, the protective role of CD8^+^ T cells in response to vaccination has raised some controversy. Kent et al. addressed this controversy and noted that the role of vaccine-elicited memory CD8^+^ T cells in directly protecting against SARS-CoV-2 infection remains unclear [20]. A study from Koutsakos et al. demonstrated that limited Spike-specific CD8^+^ T cell responses were detectable in vaccinees in the first 7–10 days following breakthrough infection of SARS-CoV-2 [21], questioning the early protective role of CD8^+^ T cells in viral control. Moreover, the level of reactive Spike-specific CD8^+^ T cells and the response of those cells elicited by the original SARS-CoV-2 vaccines against other variants of concern remain variable, with a decreased reactivity against B.1.351 and CAL20.C SARS-CoV-2 variants [22]. To harness the protective abilities of CD8^+^ T cells in developing effective immunity against all variants of SARS-CoV-2 and address current contradictions, it is crucial to understand their role in the lymph nodes (LNs), where these cells may first encounter the vaccine/Spike protein. However, the current knowledge of CD8^+^ T cells in response to vaccination in the LNs is limited. This review aims to delineate the current understanding of the protective role of CD8^+^ T cells in virologic control, their response to SARS-CoV-2 mRNA vaccines, and role of CD8^+^ T cells in the LNs.

## 2. Mode of Action of Vaccine in Activating Immune Cells

The prophylactic function of vaccination works by stimulating the immune system to produce a systemic response by mimicking the natural infection with a small fragment of pathogen or attenuated pathogen. The mRNA vaccines encoding Spike protein are encapsulated in a LNP and enter host cells by endocytosis [23]. LNPs containing ionizable cationic lipids not only assist in protecting the mRNA from lysosomal degradation and disruption but also disrupt the stable phospholipid bilayer structure of the endosomal membrane, resulting in the release of mRNA from the endosome to reach the cytoplasm [23,24]. SARS-CoV-2 Spike protein is produced via translation of mRNA by the ribosome, which is then processed by the proteasome into peptide fragments in the host APCs [23]. The antigenic peptides are then presented on the major histocompatibility complex (MHC) class I molecule on the cell surface, which in turn is recognized by Spike-specific CD8^+^ T cells (Figure 1a) [23,25]. In addition to endogenous antigen-presentation, the translated proteins can be secreted into the extracellular environment and then taken up and processed by other APCs [23,25]. Common APCs include dendritic cells (DCs), B cells, and macrophages [26]. APCs migrate to lymphoid tissue where CD4^+^ T cells recognize the peptide-MHC class II complex presented on the surface of APCs [16,27]. Activated CD4^+^ T cells differentiate into effector helper T cells, which play a versatile role in the adaptive immune response. Specifically, type 1 CD4^+^ helper T cells (Th1) secrete Interleukin 2 (IL-2) and Interferon-gamma (IFN-*γ*), which play a critical role in helping CD8^+^ T cell activation and exhibition of effector function [16,28]. In a recent study by Gressier et al., CD4^+^ T cells were shown to calibrate APCs via interferon-*a*/interferon-*b* (IFN-*a*/*b*) leading to the upregulation of transcriptional regulators on APCs [29]. This upregulation prepares APCs for CD4^+^ T cell assistance through interaction of CD40-CD40 ligand (CD40L), facilitating the interaction between APCs and CD8^+^ T cells, which ultimately results in an antiviral CD8^+^ T cell response [29]. Moreover, this study showed that for SARS-CoV-2 infection, the consolidation of IFN-*a*/*b* and CD40 signals in APCs were correlated with virus-specific CD8^+^ T cells and milder disease [29]. 

The LNs are vital secondary lymphoid organs located throughout the body where a number of immunological events are initiated. The antigen in the lymphatic vessels is recognized and then picked up by DCs, followed by the internalizing and processing of antigen by DCs. The LN fibroblastic reticular cells and high endothelial venule cells release C-C motif chemokine ligand 21 (CCL21) and CCL19, which guide DCs to enter the LNs and reach the T cell-zone [30]. Naïve CD4^+^ T cells undergo activation and differentiation into CD4^+^ follicular helper T (Tfh) cells upon encountering antigen bearing DCs [31]. These Tfh expand and proliferate in the draining LN [32]. Furthermore, the expression of CXC chemokine receptor type 5 (CXCR5) drives Tfh to migrate to the border of B cell follicles and undergo further differentiation [33]. Tfh promotes the activation of B cells and facilitates them to differentiate into antibody-secreting plasma cells and memory B cells in the germinal center (GC) following T-B cell interactions [34,35]. 

## 3. CD8^+^ T Cell Development and Cytolytic Mechanisms in Response to Viral Infection

NAbs often steal the limelight of immune responses triggered by viral infections and vaccinations. Nevertheless, CD8^+^ T cells may play a significant role in adaptive immunity against viral infections and vaccinations. The generation of CD8^+^ T cells is a lengthy process associated with many maturation and selection processes. Hematopoietic stem cells are generated and maturated into committed lymphoid progenitors in the bone marrow, and then these cells migrate to the thymus [36,37]. In the thymus, these cells differentiate into early committed T cells that lack T cell receptor (TCR) expression and are referred to as double-negative (DN) thymocytes due to their lack of CD4 and CD8 expression. There are four distinct stages of DN thymocyte development, following which DN cells develop into double-positive (DP) cells with expression of CD4 and CD8 [36,38]. TCR development occurs during the latter DN stages and the early DP stage, progressing from a non-rearranged pre-TCR a-chain and rearranged TCR b-chain to a complete TCR with rearranged and paired a- and b-chains [36,37]. The DP cells interact with self-peptides presented on MHC class I and II molecules on cortical epithelial cells [36,39]. The selection process allows the survival of cells expressing TCRs with moderate affinity that are not hypersensitive to self-peptides [36,40]. Thymocytes expressing TCRs that bind to self-peptide-MHC class I complex become CD8^+^ T cells, while those binding to self-peptide-MHC class II complex become CD4^+^ T cells [36]. Subsequently, these cells are released from the thymic medulla and migrate to peripheral lymphoid sites [36]. Upon invasion of a pathogen, the TCRs of the CD8^+^ T cells are utilized to recognize and interact with the antigen peptide fragments presented on MHC class I molecules on the surface of APCs, and this is followed by activation of the CD8^+^ T cells [41]. In response to viral infections, CD8^+^ T cells exert a predominant effector function via two distinct killing mechanisms. The first cell-killing pathway lyses infected cells by secreting effector molecules. Effector molecules involved in this lytic pathway include the pore-forming protein perforin (PFN), serine protease granzyme B (GzmB), and IFN-*γ* (Figure 1b) [42]. PFN and GzmB are commonly found in the cytoplasmic granules of CD8^+^ T cells [43,44]. Following the release of PFN from the CTL, the membranes of infected cells are disrupted due to the penetration of PFN [44,45,46]. The pores formed by PFN on the plasma membrane allow the entrance of GzmB into the cytosol, leading to cell apoptosis via signalling cascades [44,46,47,48]. The motility and cytotoxicity of CD8^+^ T cells are substantially enhanced by IFN-*γ* [49]. Furthermore, IFN-*γ* plays a vital role in the recruitment of effector CD8^+^ T cells, thereby exerting a significant influence on the immune response [50]. Tumor necrosis factor (TNF) is also secreted by effector CD8^+^ T cells (Figure 1b) and promotes apoptosis of virus-infected cells [51]. In addition to the perforin-mediated lytic pathway, CD8^+^ T cells can also induce the death of target cells through the interaction of Fas receptor and Fas ligand (FasL) (Figure 1b). The Fas receptor, also known as CD95 or Apo-1, is a surface receptor that is commonly expressed on many cell types, including T cells, B cells, and other non-immune cells, such as epithelial and endothelial cells [52,53,54]. FasL is a ligand expressed on the surface of CD8^+^ T cells that binds to the Fas receptor on the target cells [55]. As a result of the Fas-FasL binding, apoptosis of the target cells is initiated via classical caspase cascades [56]. In the context of SARS-CoV-2 infection, the functional profile of CD8^+^ T cells has been characterized in patients with severe infection. Patients with severe SARS-CoV-2 infection exhibited elevated production of GzmB and IFN-*γ* in comparison to healthy controls, while TNF production remained comparable [57,58]. Notably, the expression of PFN was inconsistent across different studies [57,58]. 

## 4. The Protective Effect of CD8^+^ T Cells in Response to SARS-CoV-2 mRNA Vaccine in Peripheral Blood

Although the role of vaccines is to induce immunity, some vaccines have suboptimal prophylactic effects, making it crucial to understand their mechanisms to induce immunity in order to optimize efficacy and prolonged protection. In the context of SARS-CoV-2, the rollout of vaccines and the regimen of prime-boost vaccination has substantially reduced mortality and morbidity [59]. As reported by Polack et al., in the phase 2 clinical trial, the protective effect elicited by vaccination was induced as early as 12 days following first dose of BNT162b2 mRNA vaccine in a large study cohort, as measured by a lower rate of occurrence of COVID-19 cases observed in the vaccine group compared to the placebo group [11]. Such protective effects may be mounted by the robust response of CD8^+^ T cells, as Spike-specific CD8^+^ T cells were detected as early as day 7 post prime vaccination [60]. However, as outlined by Painter et al., the response of CD8^+^ T cells induced by vaccine is more gradual and the magnitude is more variable than the CD4^+^ T cell response [61]. The inconsistency between studies may be attributed to the different assays used to identify Spike-specific CD8^+^ T cells, or differences in Human Leukocyte Antigen (HLA)-restricted epitopes used between study cohorts [62]. Unlike the T cell responses, NAbs elicited by vaccination were only detected 21 days after the first dose of vaccination [18]. Notably, vigorous expansions of Spike-specific CD8^+^ T cells were elicited by the vaccine with an 11-fold increase after the first dose compared to baseline, with further expansion after the second dose [63]. Aligned with the previous studies, the study from Oberhardt et al. utilized tetramer staining and successfully observed that Spike-specific CD8^+^ T cells were detected in the periphery at days 6–8 after initial or prime vaccination and peaked at days 9–12 following initial or the prime vaccination [19]. Moreover, those cells differentiated into effector CD8^+^ T cells with increased expression of T-BET, TOX, and CD39, which vigorously expanded after boost vaccination [19]. Aside from the immunophenotyping markers that have been referred to, the Spike-specific CD8^+^ T cells also displayed activation markers, such as CD69, CD154, CD137, and CD38, as well as the proliferating marker Ki-67 [60]. In addition to the massive expansion, those cells exhibited an effector function by producing IFN-*γ* and TNF, and their effector capacity was induced early after prime vaccine and remained stable after subsequent boost vaccinations [19,60]. The significance of Spike-specific CD8^+^ T cells in the containment of SARS-CoV-2 is highlighted by compelling evidence in other vaccine studies that are not mRNA-based. Pardieck et al. used a transgenic mouse model of SARS-CoV-2 infection to demonstrate that CD8^+^ T cells responding, in the absence of NAbs, to a synthetic long peptide (SLP)-based vaccine, containing a single CD8^+^ T cell epitope, provided protection against lethal SARS-CoV-2 infection in mice [64]. However, full protection was only provided after the third vaccination [64]. Liu et al. observed higher viral loads in the respiratory tract of adenovirus vector-based vaccine Ad26.COV2.S vaccinated macaques that underwent CD8^+^ T cell depletion prior to challenge with the SARS-CoV-2 virus [65]. However, the protective effect of CD8^+^ T cells elicited by mRNA vaccine that compensates for lack of humoral immunity remains to be explored. Furthermore, the protective effect of immunity induced by current mRNA vaccines against SARS-CoV-2 in humans has been observed to be long lasting and persistent. As demonstrated by Mateus et al., mRNA vaccine induced NAbs and Spike-specific memory CD8^+^ T cells remained detectable in vaccinees at least 6 months post second vaccination [66]. Similarly, Ozbay Kurt et al. observed that Spike-specific CD8^+^ T cells could be detected 4–6 months after the second vaccination [67]. However, a gradual decrease in the frequency of these cells was noted 12 weeks after the booster vaccination [67]. In the same study, Spike-specific CD8^+^ T cells showed an ability to be stimulated to expand again after third dose followed by a contraction observed a few months later [67]. 

## 5. CD8^+^ T Cells and SARS-CoV-2 Epitopes

Over the past few years, a number of studies have been undertaken to demonstrate that Spike-specific CD8^+^ T cells can ameliorate recovery from SARS-CoV-2 infection. Identification of dominant HLA types and SARS-CoV-2 epitopes is the fundamental task before conducting such research. An epitope is an immunogenic domain of the antigen, which binds to the MHC class I and class II molecules (HLA on human cells) and triggers an immune response by CD8^+^ and CD4^+^ T cells, respectively [68]. Notably, the CD8^+^ T cell specific epitopes are shorter than CD4^+^ T cell specific epitopes, composed of only 8 to 11 amino acid residues [69]. Hence, the activation and effector function of CD8^+^ T cells can be easily compromised by a single mutation on the epitope [70]. 

To date, scientists have identified over 1000 SARS-CoV-2 CD8^+^ T cell epitopes with the utilization of peptide stimulation and peptide-tetramer staining [69]. There are ten HLA class I alleles dominant for SARS-CoV-2 CD8^+^ T cell epitopes, including HLA-A*01:01, -A*02:01, -A*03:01, -A*11:01, -A*24:02, -B*07:02, -B*08:01, -B*15:01, -B*40:01, and -C*07:02 [71]. In the study by Nelde et al., 81% of pre-pandemic donors had T cell responses to cross-reactive SARS-CoV-2 epitope compositions, which was shown to be similar to common cold coronavirus [71]. Interestingly, the HLA-B*07:02-restricted nucleoprotein (N)_105-113_ epitope (B7/N_105_) is more dominantly targeted by CD8^+^ T cells [72]. Nguyen et al. identified SARS-CoV-2 specific CD8^+^ T cells in pre-pandemic populations and patients infected with SARS-CoV-2 by using peptide-HLA tetramers. In this study, CD8^+^ T cells specific for B7/N_105_ were detected in higher frequencies in pre-pandemic unexposed individuals and COVID-19 patients than CD8^+^ T cells specific for three subdominant epitopes [73]. Notably, B7/N_105_ tetramer-specific CD8^+^ T cells displayed a naïve phenotype in pre-pandemic populations [73]. However, whether individuals with the HLA-B*07:02 allele are less susceptible to severe disease is not clear from this study. It is worth considering that this study was conducted with a small study cohort of 61 subjects within the local area. In fact, another research study analyzing 4361 subjects indicated that there was no correlation between HLA types and SARS-CoV-2 susceptibility [74]. What is even more intriguing is that specific alleles could induce a substantially greater T cell response compared to others. Gao et al. pointed out that although both HLA-A*02:01- and HLA-B*40:01-restricted Spike epitopes induced a more than 35-fold rise in the frequency of Spike-specific CD8^+^ T cells, the magnitude of the HLA-B*40:01-specific T cell response was inferior to that of the HLA-A*02:01-specific response [60].

## 6. Vaccine-Elicited CD8^+^ Memory T Cells

The key to long-term protection from vaccination is the establishment of immunological memory. Immunological memory reflects the development of memory subsets within CD8^+^ T cells that are specific to particular antigens, allowing for faster recognition and stronger recall immune responses upon re-exposure to the antigens [75,76]. Apart from NAbs, many vaccines also aim to generate subsets of memory CD4^+^ and CD8^+^ T cells that persist to provide durable protection. 

There are four main types of memory CD8^+^ T cells identified in previous studies, including central memory T cells (T_CM_; CD45RA^−^ CCR7^+^ CD27^+^), effector memory T cells (T_EM_; CD45RA^−^ CCR7^−^ CD27^−^), stem cell memory T cells (T_SCM_; CD45RA^+^ CCR7^+^ CD27^+^ CD95^+^), and terminally differentiated effector memory CD45RA^+^ T cells (T_EMRA_; CD45RA^+^ CCR7^−^ CD27^−^) [19,63,75,77,78,79,80,81]. Vaccine-induced memory CD8^+^ T cells circulating in the periphery can be rapidly induced as soon as one month after prime-boost vaccination. In the study by Papadopoulou et al., they used specificity and cytotoxicity assays to show that the vaccine-induced memory CD8^+^ T cells can be rapidly activated and exert a protective function following the challenge of the virus even 8 months after first vaccination [82]. A Spike-specific memory precursor pool of CD8^+^ T cells expressing memory markers CD127 and T cell factor 1 (TCF1) was found in circulation at day 6–8 after the first vaccine dose [19]. As reported by Kondo et al., a population of antigen-specific CD8^+^ T cells expressing early activation markers HLA-DR and CD38 was detected 21 days after boost, which is an indicator for the generation of memory CD8^+^ T cell subsets [83]. Notably, those memory CD8^+^ T cells were composed of a considerable proportion of T_EM_ [83]. This result aligns with the studies of Oberhardt et al. and Gao et al., where they also identified a high frequency of CD8^+^ T_EM_ cells in vaccinated subjects at the early timepoints after prime-boost vaccination [19,60]. Rapid and potent immune response recall is the main characteristic of T_EM_. These T_EM_ populations progressively contracted with an expansion of T_CM_ cells during the later phase of the immune response to vaccination [60]. These memory subsets form a stable memory cell pool that can promptly recognize and eliminate infected cells when virus is re-encountered. In addition to the identification of a high proportion of T_EM_ cells post prime-boost vaccination, a considerable proportion of T_EMRA_ were identified in the periphery of vaccinees after 2 months post boost vaccination [83]. T_EMRA_ are the most terminally differentiated effector memory CD8^+^ T cells that re-express CD45RA and have potency of effector function [19]. However, the SARS-CoV-2 specific T_EMRA_ have been described as non-typical T_EMRA_. In the study by Neidleman et al., the phenotype of SARS-CoV-2 specific T_EMRA_ was compared to T_EMRA_ specific for Cytomegalovirus (CMV) [80]. The majority of SARS-CoV-2 specific CD8^+^ T_EMRA_ cells expressed high levels of CD27 and CD28 compared to the CMV-specific T_EMRA_ cells, indicating that these cells are less terminally differentiated [80]. Interestingly, as the phenotypic characterization of SARS-CoV-2, specific CD8^+^ T_EMRA_ cells identified in this study showed similarity to previously reported Epstein-Barr virus (EBV)-specific CD8^+^ T_EMRA_ cells [84]. Neidleman et al. proposed that the SARS-CoV-2 specific CD8^+^ T_EMRA_ cells are cytotoxic and long-lived [80], but further research will be needed to validate this hypothesis. The T_SCM_ subset has been suggested to be associated with long-term T cell immunity, and a SARS-CoV-2-specific T_SCM_ cell subset persistent across a 6-month study has been identified by Guerrera et al. [63]. More importantly, Reinscheid et al. showed that the T_SCM_ pool was expanded after the first vaccine dose and remained stable with subsequent vaccinations [81]. Interestingly, a high proportion of SARS-CoV-2 specific T_SCM_ was also detected in SARS-CoV-2 infected subjects a few months after symptom onset [85]. This study also emphasized the ability of T_SCM_ cells to self-renew and differentiate into diverse memory subsets, including T_EM_, T_EMRA_, and T_CM_ [85]. Additionally, T_SCM_ cells have been previously studied in other vaccine contexts. In the study by Fuertes Marraco et al., they reported the induction of a population of yellow fever-specific CD8^+^ T cells with a naïve-like phenotype (CD45RA^+^ CCR7^+^) after yellow fever vaccination [86]. With further investigation, they found that this population expressed CD58, CD95, and CXCR3, which was phenotypically distinct from “bona fide” naïve cells [86]. Remarkably, this T_SCM_ population was sustained for more than 25 years [86].

It is interesting to note that the distribution of Spike-specific memory CD8^+^ T cell subsets elicited by vaccination and natural infection were distinct even at the same time point post vaccination/infection. In the study by Oberhardt et al., T_CM_ and early differentiated (T_ED_; CD45RA^+^ CCR7^+^ CD27^+^ CD11a^+^ CXCR3^+^) CD8^+^ T cell subsets specific for SARS-CoV-2 spike peptide-loaded HLA class I tetramers (HLA-A*01-restricted Spike (S)_865,_ (HLA-A*01/S_865_) and HLA-A*02/S_269_) and transitional memory (T_TM_; CD45RA^+^CCR7^−^ CD27^+^) CD8^+^ T cells specific for HLA-A*03/S_378_ were more dominant in convalescent individuals after 80 days of natural infection, while T_EM_ were more prevalent in vaccinees at the same timepoint [19]. The differences in memory CD8^+^ T cell pools induced by vaccination and natural infection may be attributed to variations in the routes of antigen exposure between infection and inoculation and the induction of only Spike proteins as antigens in current mRNA vaccines. More importantly, SARS-CoV-2-specific T cell responses elicited by natural infection were shown to be more durable than vaccine-induced responses. Two other studies reported that SARS-CoV-2 specific CD4^+^ and CD8^+^ T cell responses remained detectable more than 10 months post infection [85,87]. The difference in memory subset composition may contribute to the more durable T cell responses in convalescent individuals. However, further research into vaccine-elicited memory CD8^+^ T subsets is required to clarify the correlation between duration of protection and memory T cell subset composition, which could inform optimization of vaccination. 

As noted earlier, there has been some debate among researchers regarding the extent to which vaccine-elicited CD8^+^ T cells protect against SARS-CoV-2 infection. This controversy was partly due to the lack of sufficient data linking the level of circulating memory CD8^+^ T cell subsets to the degree of protection conferred against infection in vaccinees [20]. According to Kent et al., this data gap can be partly attributed to the absence of a standardized assay for measuring T cell responses [20]. Utilizing tetramers or multimers has facilitated the successful identification of antigen-specific CD8^+^ T cells, whilst its limitation cannot be neglected. Since this method heavily relies on the binding affinity between the peptide-tetramer complex and TCR, it sometimes fails to entirely detect the full range of functional T cell clonotypes, resulting in an underestimation of the antigen-specific CD8^+^ T cell population [88]. Therefore, the protocol for using tetramers or multimers should be carefully optimized before the commencement of research. Alternatively, activation induced marker (AIM) assays that utilize a wider pool of overlapping peptides are an effective and sensitive tool for the identification of antigen-specific CD8^+^ T cells [89,90].

## 7. CD8^+^ T Cells in Lymph Nodes

LNs are an important site for initiating immune responses and play a crucial role in controlling viral infections. The role of vaccine-induced CD8^+^ T cells providing sterilizing immunity in the LNs has been investigated in previous studies. In a simian immunodeficiency virus (SIV) study, vaccine-induced CD8^+^ T cells significantly reduced the viral load in the LNs in vaccinated rhesus macaques following challenge of SIV mac239 virus [91]. Interestingly, in a recombinant adenovirus vector vaccine study in mice, CD8^+^ T cells rapidly expanded in the draining LNs, secreted IFN-*γ* in response to ex vivo stimulation and migrated through the lymphatics following vaccination [92]. In the context of respiratory viruses, a recent study has suggested that the current mRNA vaccines may not fully utilize the protective function of CD8^+^ T cells in controlling viral infection [93]. One possible reason for this is that the vaccines do not induce a strong response of memory CD8^+^ T cells that reside in the respiratory barrier site where the virus initially invades [94]. Although such a population has a high potential to eradicate virus to prevent local infection [95], its induction requires activation of effector CD8^+^ T cells in the LNs and their migration to the peripheral site [93]. Whilst LN CD8^+^ T cells have been investigated in the context of other vaccines, the response of LN CD8^+^ T cells elicited by SARS-CoV-2 mRNA vaccines remains unclear. 

Naïve CD8^+^ T cells in the LN, such as circulating CD8^+^ T cells, become activated and differentiate upon recognition of the antigen presented by APCs (Figure 2) [96]. Recent studies have identified a novel CD8^+^ T cell subset in the LNs. These cells expressing CXCR5 are known as follicular cytotoxic T cells (Tfc). CD8^+^ T cells that are CXCR5^+^ TCF1^+^ Tim3^−^ have been identified as Tfc [97]. Analogous to the function of CD4^+^ Tfh, Tfc have the ability to migrate to the B-cell follicle guided by CXCR5 and assist B cells during GC reactions by secreting Interleukin 21 (IL-21) [98,99]. In addition to IL-21, Tfc also secrete IL-2, IL-4, IFN-*γ*, TNF-a, GzmB, and PFN [100,101]. Virus-specific Tfc have been identified in the blood and LNs of patients with chronic human immunodeficiency virus (HIV) infection [102]. The majority of Tfc are localized in the secondary lymphoid organs, including the LNs, spleen, and tonsils, while a very small proportion of Tfc are found in the periphery [103,104,105]. Previous studies claimed that Tfc exhibit a distinct memory phenotype (CD45RO, CD69, CD127, CD62L) [97,103], whilst these cells have a lower expression of cytotoxic transcripts (*GZMA*, *GZMB*, *PRF1* and *IFNG*) compared to CXCX5^−^ cells [106,107]. The lack of correlation between high cytokine secretion and low cytotoxic gene expression may be attributed to the disconnect between protein expression and gene expression, which could arise from differences in turnover time. However, further investigation is still needed for validation. Notably, these cells are less functionally exhausted as they do not express inhibitory receptors (Tim-3 and 2B4) [106] and express a lower level of CCR7 than CXCR5^−^ cells [100], whilst the expression of PD-1 is variable between studies [100,106,107,108]. These cells are capable of self-renewal and rapid proliferation [107]. A number of studies have shown that the differentiation and function of Tfc are regulated by various transcription factors, including TCF1, Bcl6, Id2, and Blimp1. In particular, the expression of TCF1 and Bcl6, which enhance memory CD8^+^ T cell formation, are found upregulated in Tfc, whilst the transcription factors Id2 and Blimp1 suppressing the expression of CXCR5 have been found to be downregulated [97,100,106,109]. Given their cytotoxic nature, Tfc act as a defender against viruses. Studies have demonstrated an inverse correlation between the frequency of circulating and LN Tfc and the viral load in the periphery, further supporting their role in antiviral defence [108,109,110,111]. Leong et al. suggested that Tfc have a vital function in surveilling and eliminating infected cells present in the B cell follicles [108]. Their study suggested that Tfc control viral infection of Tfh in mice infected with a virus sharing certain characteristics with HIV, specifically LCMV [108]. Furthermore, in the context of HIV, Tfc are capable to infiltrate the GC to eradicate HIV-infected cells [101]. In addition to viral control, Tfc also play a role in eliminating malignant cancer cells [112] and exerting helper function in inflammation and autoimmune disease [106]. Given the fact that most of the research on Tfc has been directed towards its association with HIV and other diseases, investigating the role and function of Tfc in response to SARS-CoV-2 and mRNA vaccines would be an intriguing area of study.

Our group and others have utilized ultrasound guided fine needle biopsy (FNB) or aspiration (FNA) to sample human LNs to quantify CD4^+^ T and B cell responses to infection [113] and vaccination [32,114], including the work in progress by our group on LN responses to SARS-CoV-2 vaccines. These investigations have yielded valuable immunological insights. For example, Law et al. used FNB samples from LNs five days post influenza vaccination to demonstrate that early expansion of GC B cells and Tfh cells occurs exclusively within the draining LNs and that circulating Tfh in peripheral blood may not be a good surrogate for studying early Tfh responses [32]. Similarly, Mudd et al. used FNA to observe a robust and persistent Tfh response in human draining LNs following vaccination with a SARS-CoV-2 mRNA vaccine [114]. As little is understood about human CD8^+^ T cell responses in the LNs, with most studies focused on peripheral blood, an exciting avenue for future investigation that could potentially inform vaccine design would be applying similar techniques to investigate the LN CD8^+^ T cell responses to SARS-CoV-2 vaccines.

**Figure 2 biology-12-01035-f002:**
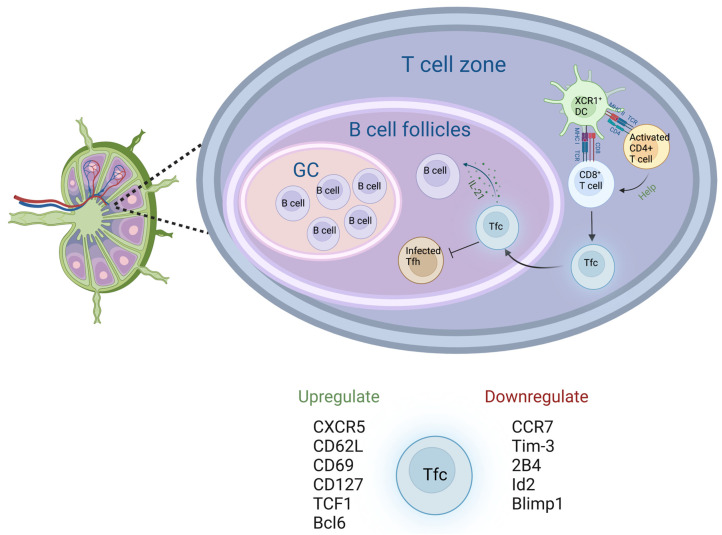
CD8^+^ T cells in LN. Naïve CD4^+^ T cells become activated, and then interact with and license cognate XC-chemokine receptor 1 (XCR1)^+^ DCs [115]. Naïve CD8^+^ T cells recognize antigen-presenting MHC class I molecule on XCR1^+^ DCs, become activated, and then differentiate into Tfc in the T cell zone. Tfc then migrate to B cell follicles and release IL-21 to help the differentiation of B cells into memory B cells and antibody-producing plasma cells in the GC. Tfc also have effector functions in eradicating infected Tfh during viral infection of HIV or LCMV. DC, dendritic cell; GC, germinal center; Tfc, follicular cytotoxic T cell; Tfh, follicular helper T cell; XCR1, XC-chemokine receptor 1. Figure was created with BioRender.

## 8. Concluding Remarks

In recent years, the development and deployment of SARS-CoV-2 mRNA vaccines has been a remarkable achievement in the global effort to combat the ongoing COVID-19 pandemic. The protective role of CD8^+^ T cells in the context of other vaccinations has been extensively explored and well-established. While there have been investigations of the SARS-CoV-2 specific CD8^+^ T cell epitopes and CD8^+^ T cell memory subsets, there are still many questions associated with the CD8^+^ T cell response to SARS-CoV-2 mRNA vaccines that need to be addressed. It is not yet known whether CD8^+^ T cells in LNs exhibit similar characteristics to circulating CD8^+^ T cells in the context of vaccination. Furthermore, the variations in memory subsets, phenotype, and TCR diversity of LN CD8^+^ T cells after the first, second, and third vaccinations, as well as in the draining and non-draining LNs, are still awaiting exploration. Moreover, vaccine-elicited responses of CD8^+^ T cells wane within a few months, which can compromise the protective efficacy. Therefore, a better understanding of the role of CD8^+^ T cells in the LNs is crucial in harnessing their protective abilities and in mounting effective and long-lasting immunity against all variants of SARS-CoV-2.

## Figures and Tables

**Figure 1 biology-12-01035-f001:**
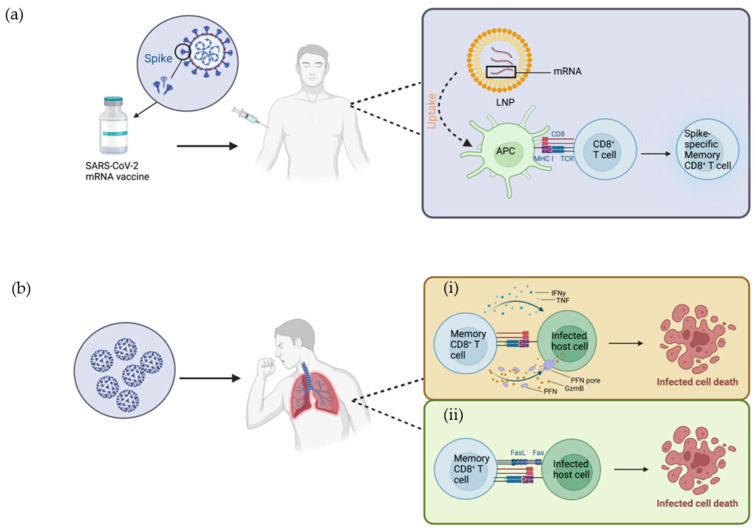
CD8^+^ T cells in response to SARS-CoV-2 mRNA vaccine and SARS-CoV-2 infection. (**a**) Response to mRNA vaccine: The SARS-CoV-2 mRNA vaccine consists of an LNP-encapsulated mRNA strand that encodes the SARS-CoV-2 Spike glycoprotein, which elicits systemic immune responses upon injection. The vaccine-elicited immune response begins with the endocytosis of the LNP and release and processing of the mRNA intracellularly, which is then processed into small peptide fragments. The peptide fragments act as an endogenous antigen and are presented via the MHC class I molecule. The peptide-MHC I complexes are recognized by the TCRs of the CD8^+^ T cells. CD8^+^ T cells become activated upon the recognition of antigen, followed by differentiation into spike-specific memory CD8^+^ T cells. (**b**) Two response pathways to infection: (i) Perforin-mediated pathway: Memory CD8^+^ T cells recognize the infected cell and release effector molecules, such as PFN, GzmB, IFN-*γ* and TNF. PFN forms pores on the plasma membrane of the target cell, allowing GzmB to enter the cytosol leading to cell apoptosis via signalling cascades. (ii) Fas-mediated pathway: Apoptosis of target cells can also be initiated via the interaction of the Fas receptor on its surface and FasL on CD8^+^ T cells. FasL, Fas ligand; GzmB, granzyme B; IFN-*γ*, Interferon-gamma; LNP, lipid nanoparticle; mRNA, messenger RNA; MHC I, major histocompatibility complex class I molecule; PFN, perforin; TCR, T cell receptor; TNF, Tumor necrosis factor. Figure was created with BioRender.

## Data Availability

Not applicable.

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
