# Peer review of "Dissecting the Protective Effect of CD8+ T Cells in Response to SARS-CoV-2 mRNA Vaccination and the Potential Link with Lymph Node CD8+ T Cells"

_biology, 2023, doi:10.3390/biology12071035_

Round 1

Reviewer 1 Report

Chen et al propose a review on CD8+ T cell immunity after mRNA-based SARS-Cov-2 vaccination. The review is starting with very basic information on the mode of action of vaccines and the development and function of effector CD8+ T cells. This is follows by a brief summary on described spike- or nucleocapsid-derived CD8+ T cell epitopes and on the subset of memory T cells that could be stimulated by vaccination. Finally, the authors introduce lymph-node resident follicular CD8+ T cell (Tfc) and how these cells could be involved in CD8+ T cell memory against SARS-Cov-2 following vaccination. This is however speculative, since no study so far has really addressed the role of this population in the context of SARS-Cov infection and/or vaccination.

The review is well written and nice to read, but it appears very basic for the immunologist readership, especially the sections 2 and 3 stay very general and do not really introduce specific aspects which could be of particular interest for the topic of SARS-Cov-2 vaccination. The part on Tfc is interesting, but since there is no data available yet in relation with SARS-Cov-2, in my opinion it does not integrate very well to the rest.

Main points:

The title is misleading, since, as we learn in the last sentence of section 8, there is currently no data available on lymph node CD8+ T cells in relation with SARS-Cov-2 vaccination (please see also above).

In section 5, lines 227-232: it is not clear what the authors means with “pre-infected” people, or do they mean “pre-pandemic”? In ref 67, tetramers were used for assessing T cell frequencies in non-infected people. The use of overlapping peptides is only possible to recall memory (or effector cells). It should be clarified whether naive or memory T cells are discussed here. Also the sentence “However, it does not mean…” is not really clear.

In section 6, lines 261-263: it sounds atypical that TEM cells “differentiate” in “TCM” cells, I don t think this is what the authors demonstrate in ref 53, please double check this.

In section 6, differentiation stages of CD8+ T cells:

TEMRA cell description. CMV specific cells are known to be very dysfunctional /exhausted and might not represent “bona fide” TEMRA cells. Do the authors have any information on the expression of CD27 on TEMRA cells specific for other viruses e.g. EBV or Flu?

Lines 286 and 291: please define what “naïve-like” and “transitional memory” T cells are.

Table 1: this table is very succinct and in the current form does not really appear essential. Also, the phenotype given for TSCM also applies for naïve cells.

Section 7 is very short and could be put together with the following section on Tfc

Minor:

2nd section (MoA of vaccines): sentences lines 90-93: vaccination may be also based on attenuated pathogen, not only on fragment(s) of it. The 2nd sentence seems strange, please check the English.

3rd section (CTL development): line 132, “fragment” should be plural. IFNg play a significant role also in recruiting further effector cells, this is missing if the aim is to describe the function of effector CD8+ T cells. Also, it is also not clear why TNF is not mentioned.

Fig 1: in the panel a), it looks like the entire virus is contained in the mRNA vaccine, replacing the dotted lines by an arrow would be better. In panel b) I would suggest removing the mask, since contamination with the virus is clearly more likely to happen when not wearing a mask. Fig 1 legend: line 154 “encodes the” and line 161 (CD8”+” should be uppercase, see also line 306)

Fig2: I would suggest adding CD4+ cells here, as they are essential for CD8+ priming and B cell activation, also the authors could show antibody production.

Lines 202-205: please check English

Line 225: “N” has not be defined before

Line 273 and 281: “were” should be replaced by “was”

Lines 284 and 347: is there a specific reason for writing some words in bold characters?

Line 291: define what A*01/S865 and A*02/S269 means

Line 349: point is missing

Sentence lines 369-370 needs to be rephrased

References: I would recommend adding an essential ref for the description of SARS-Cov 2 epitopes i.e. Nelde et al. PMID: 32999467. In the discussion on the decrease of T cell immunity after vaccination (lines 207 and following), it would be interesting to integrate the work of Kurt et al. which dissects T cell reactivity after booster vaccinations (PMID: 36311732).

Ref 53: There is a formatting problem (the title is not correct).

English is overall very clear, only a couple of minor comments (see above)

Reviewer 2 Report

In this manuscript, the authors explore the current understanding of the protective role of CD8+ T cells, their response to SARS-CoV-2 mRNA vaccines, and their functions within the lymph nodes. Overall, the manuscript is well-structured and written. Just one very minor suggestion, re-position (i) and (ii) in Figure 1.

Reviewer 3 Report

The review is a review of studies on the immune response to coronavirus COVID-19 with a focus calling attention to the role of CD8 T cells as an important component of protective immunity. The review is written for the broader audience of the research community interested in development of protective immunity against COVID-19 but not directly involved in research on the immune response to COVID-19. This is the main contribution, especially in highlighting studies showing CD8 CTL play a role in protection. The manuscript needs to be revised in respect to development of CD8 CTL. The description on the provision of help to CD8 by CD4 T cells is inaccurate. The general thought has been that CD4 T cells provide help to CD8 T cells indirectly through licensing APC. Recent studies indicate that the actual interaction occurs at the interface between APC, CD4 and CD8 T cells. The review should stress this is an area that needs to be investigated. It should also be emphasized that the immune response always involves antibody and CD8 T cell responses. This is where the immune response to pathogens needs to be clarified. Also, the encapsulated mRNA is taken up by various stomal cells that make and secrete the Ag. The Ag is taken up by APC that process and present the Ags through MHC I and II on the APC. A little more clarification will make this a more relevant contribution.

Round 2

Reviewer 1 Report

Thank you for the revised manuscript and for addressing my comments. I have very few points left.

Table 1: This table still looks poorly informative, for example, naïve T cells, stem cell memory and early differentiated T cells are listed with the same phenotype. So the phenotypes given appear incomplete and need to be more precise. Please also add refs for the definition of the phenotypes.

Fig2: thanks for adding a CD4+ T cell, however it would be more exact to show it interacting with the APC (as done for the CD8+ T cell)

Minor points:

Lines 144-145 "self-antigen" sounds strange, I would replace by "self-peptide"

Lines 229-232: I think that the sentence is still not very clear, maybe rephrase for "Liu et al. observed higher viral loads in the respiratory tract…."

New Lines 260- 267: "at higher frequencies in pre-pandemic populations and at higher frequencies during acute and convalescent COVID-19". Please clarify was higher frequencies means (higher than what?)

Line 354: multimers not multi-mers

Line 424:"These investigations have yielded valuable immunological insights" the authors need to at least mention shortly which new findings were made

English is good, just a few comments above
